# *Kocuria* Species Infections in Humans—A Narrative Review

**DOI:** 10.3390/microorganisms11092362

**Published:** 2023-09-21

**Authors:** Afroditi Ziogou, Ilias Giannakodimos, Alexios Giannakodimos, Stella Baliou, Petros Ioannou

**Affiliations:** 1School of Medicine, National and Kapodistrian University of Athens, 11527 Athens, Greece; 2School of Medicine, University of Crete, 71003 Heraklion, Greece

**Keywords:** *Kocuria*, bacteremia, bloodstream infection, endocarditis, peritonitis, endophthalmitis, skin and soft tissue infection

## Abstract

*Kocuria* species are catalase-positive and coagulase-negative Gram-positive coccoid bacteria that belong to the family Micrococcaceae, order Actinomycetales, and class Actinobacteria. Even though they may be relatively rare, they have been increasingly reported as the causes of human infections lately. The present study aims to review all published cases of *Kocuria* spp. infections in humans reporting data on epidemiology, microbiology, antimicrobial susceptibility, antimicrobial treatment, and mortality. A narrative review was performed based on a search of Pubmed and Scopus databases in the literature. In total, 73 studies provided data on 102 patients with *Kocuria* spp. infections. The mean age of patients was 47 years, and 68.3% were male. The most common types of infection were bacteremia (36.3%), skin and soft tissue infection (18.6%), endophthalmitis (15.7%), infective endocarditis (13.7%), and peritonitis (11.8%), most commonly peritoneal–dialysis-associated. The most frequently isolated species was *K. kristinae *(46.1%), and antimicrobial resistance was lower for vancomycin (7%) and tetracyclines (6.7%). Vancomycin (47%), cephalosporins (39.6%), and quinolones (36.6%) were the most commonly used antimicrobials. The empirical antimicrobial treatment of *Kocuria* spp. infections should include vancomycin as long as antimicrobial susceptibility results are pending. The infection outcome mainly depends on the type of infection and is higher for infective endocarditis. Endophthalmitis is associated with increased rates of low visual acuity after treatment.

## 1. Introduction

*Kocuria* species are catalase-positive and coagulase-negative Gram-positive coccoid bacteria that belong to the family Micrococcaceae, order Actinomycetales, and class Actinobacteria. They can be identified as tetrads or irregular clusters. *Kocuria* comprises 18 species [1,2,3]. These microorganisms were previously classified as *Micrococcus*, but after the phylogenetic analyses of Stackebrant et al., taxonomic changes were performed based on 16s rDNA sequences as well as the amino acid composition of peptidoglycan [2,3]. These microorganisms are ubiquitous and can be isolated from various sources, such as the skin, the flora of mammals, and soil. These microorganisms usually colonize the skin, mucosae, and oropharynx in humans. Among the 18 species of *Kocuria*, 5 are known to be opportunistic pathogens, more commonly in immunocompromised patients or patients that have severe underlying debilitating comorbidities [1,2]. Due to difficulties in pathogen identification and the phenotypical similarity between *Kocuria* spp. and coagulase-negative staphylococci, misidentification between these two genera may have led to the underdiagnosis of *Kocuria* spp. [1,4]. Advanced microbiological techniques, such as 16rRNA, are required for accurate identification [5]. Currently, no official treatment guidelines exist for *Kocuria* infections; most cases are initially treated empirically until antimicrobial susceptibility results are available. The mortality of these infections is generally low and mainly depends on the site of the infection as well as the patient’s comorbidities.

The primary aim of the present study was to review all published cases of *Kocuria* spp. infections in humans reporting data on epidemiology and mortality. Another objective is to describe these human infections’ microbiology, antimicrobial susceptibility, and antimicrobial treatment. This review constitutes an attempt to fill certain knowledge gaps, including predisposing factors and treatment options, and to enhance the limited data available in the literature for this pathogen.

## 2. Materials and Methods

This narrative review was performed to export data on *Kocuria* species infections in humans. For the conduction of this review, three investigators (A.G., A.Z., I.G.) independently searched PubMed/Medline and Scopus databases for eligible articles reporting on “*Kocuria*” until 20 May 2023. Any dispute was resolved by the intervention of a senior investigator (P.I.). This narrative review included case reports and case series providing information on the epidemiology, microbiology, treatment, and outcomes of *Kocuria* infections in humans. Only articles written in the English language were accepted. Reviews, systematic reviews, retrospective studies, and letters to the editor were excluded. Articles with no access to original data and studies referring to animal reports had to be excluded. Moreover, studies with insufficient data and articles without information on patients’ mortality and epidemiology were excluded as irrelevant. The remaining articles’ references were also searched to assess potential studies following the snowball procedure.

Three investigators (I.G., A.G., and A.Z.) extracted information from all the eligible studies included in this narrative review using a pre-defined template. Data concerning age, epidemiology characteristics, the site of infection, the microbiology of species, susceptibility methods, antibiotic information, treatment, and the outcomes of *Kocuria* subspecies infections in humans were accumulated and used for the results’ extraction.

## 3. Results

### 3.1. Included Studies’ Characteristics

The search of the literature retrieved 503 studies. After the exclusion of the duplicate articles, the record screening, and the snowball procedure, only 73 studies met the inclusion criteria. These 73 studies provided data on 102 patients, and these data were included in the analysis [4,6,7,8,9,10,11,12,13,14,15,16,17,18,19,20,21,22,23,24,25,26,27,28,29,30,31,32,33,34,35,36,37,38,39,40,41,42,43,44,45,46,47,48,49,50,51,52,53,54,55,56,57,58,59,60,61,62,63,64,65,66,67,68,69,70,71,72,73,74,75,76,77]. A flow diagram of the selection process is shown in Figure 1. Among them, 39 studies were conducted in Asia, 20 studies in Europe, 13 studies in North and South America, and one study in Africa. There were 64 case reports and 9 case series. Figure 2 shows a graphical representation of the geographical distribution of these published cases.

### 3.2. Epidemiology of Kocuria spp. Infections in General

The ages of the patients ranged from 3 months to 90 years. The mean age was 47 years, and 69 out of 101 patients (68.3%) were male. Active solid or hematologic malignancy was the most common underlying disease in 21.6% of patients, while 13.4% of patients were being treated with chemotherapy at the time of diagnosis and infection. A central venous catheter (CVC) was present in 18.4% of patients, and 13.1% were on total parenteral nutrition (TPN). Patients with end-stage renal disease (ESRD) on peritoneal dialysis made up 10.8% of all cohorts, while 2.9% had ESRD but were on hemodialysis. Moreover, 4.8% of patients had bad teeth hygiene, and 3.7% were intravenous drug users (IVDU). Recent antimicrobial use (within the last three months) was noted in 13% of patients.

### 3.3. Microbiology and Antimicrobial Resistance of Kocuria spp. Infections in General

*Kocuria* spp. was isolated from the blood in 34.3% of patients, in pus cultures in 21.6%, in vitreous fluid in 15.7%, in peritoneal fluid in 11.8%, in corneal tissue in 3.9%, in cerebrospinal fluid (CSF) in 2.9%, in sputum in 2.9%, and brain tissue, in canaliculus fluid, in bone culture, in nail lesions, in an implantable cardioverter defibrillator (ICD) lead, in nasopharyngeal culture, and urine culture in 1%. The most commonly identified species was *K kristinae* at 46.1%, *K. rosae* at 21.6%, *K. varians* at 8.8%, *K. rhizophila* at 7.8%, *K. marina* at 4.9%, *K. koreensis* in 2%, *K. palustris* in 2%, *K. salsicia* in 2%, *K. arsenatis* in 1%, and *K. massiliensis* in 1%. In 2.9%, the species was not specified. For the identification of *Kocuria* spp., VITEK 2 (BioMerieux Inc., Durham, NC, USA) was used in 60%, 16s rRNA in 21.3%, and 16s rRNA along with the matrix-assisted laser desorption/ionization time-of-flight mass spectrometry (MALDI-TOF MS) in 8%, MALDI-TOF MS in 8%, the API Staph System in 1.3%, and the BD BBL Crystal GP system in 1.3%.

Regarding antimicrobial resistance, several methods were applied for its examination, depending on the laboratory; the E-test or the disk diffusion method constituted the most commonly used methods. Among the studies providing data, macrolide resistance was noted in 52.4%, penicillin resistance in 51%, clindamycin resistance in 44.7%, aminoglycoside resistance in 36.2%, trimethoprim-sulfamethoxazole resistance in 30.4%, quinolone resistance in 28.4%, cephalosporin resistance in 17.9%, rifampicin resistance 16.7%, vancomycin resistance in 7%, and tetracycline resistance in 6.7%.

### 3.4. Clinical Presentation of Kocuria spp. Infections in General

The most common infections caused by *Kocuria* spp. were bacteremia in 36.3%, skin and soft tissue infection (SSTI) in 18.6%, infective endocarditis (IE) in 13.7%, endophthalmitis in 15.7%, peritonitis (most commonly in patients on peritoneal dialysis) in 11.7%, an upper respiratory tract infection in 5.9%, central nervous system infection in 5.9%, bone and joint infection in 4.9%, keratitis in 4.9%, lower respiratory tract infection in 2%, and urinary tract infection in 2%. Fever was present in 43.3% of patients, organ dysfunction in 19.5% (more commonly, heart failure in 8%), sepsis in 14.4%, and shock in 5.7%.

### 3.5. Treatment and Outcomes of Kocuria spp. Infections in General

The most commonly used antimicrobials for the treatment of *Kocuria* spp. infections were vancomycin in 47%, cephalosporins in 39.6%, quinolones in 36.6%, linezolid in 17%, aminoglycosides in 14%, penicillin in 7.9%, aminopenicillin in 6.9%, clindamycin in 5.9%, carbapenems in 5%, and daptomycin in 2%. Surgery was performed along with antimicrobials in 36.6%. Overall mortality was 5.9%, and mortality that was attributed directly to the infection was 4.9%.

### 3.6. Bacteremia Due to Kocuria spp.

The mean age of patients with bacteremia caused by *Kocuria* spp. was 42.5 years, and 61.1% were male. A CVC was present in 45.7% of patients, and 36.1% were on TPN. The most common clinical presentation of bacteremia by *Kocuria* spp. was fever in 81.1%, sepsis in 32.4%, organ dysfunction in 26.5%, and shock in 13.9%. IE was present in 37.8% of patients. Overall mortality was 8.1%, and infection-specific mortality was 5.4%. Table 1 shows the characteristics that are among the most common clinical presentations of infections by *Kocuria* spp.

### 3.7. Infective Endocarditis Due to Kocuria spp.

The mean age of patients with IE caused by *Kocuria* spp. was 58.2 years, and 85.7% were male. Bad teeth hygiene or recent dental work was noted in 21.4%, 14.3% were IVDU, 14.3% had a central venous catheter, 14.3% of patients had a recent cardiac surgery (within three months), 7.1% had an implantable cardiac device, 7.1% had congenital heart disease, and 7.1% had a history of rheumatic fever. No patient had a prosthetic cardiac valve. The aortic valve was infected in 50%, the mitral valve in 50%, the pulmonary valve in 7.1%, and an implantable cardiac device was present in 7.1%, while multiple valves were infected in 14.3%. The diagnosis was facilitated by transthoracic echocardiography in 78.6%, transesophageal echocardiography in 14.3%, and culture of an extracted intracardiac lead in 7.1%. The most common clinical presentation of IE by *Kocuria* spp. was fever in 78.6%, organ dysfunction in 50% (more commonly heart failure), embolic phenomena in 35.7%, sepsis in 28.6%, shock in 28.6%, mycotic aneurysm in 14.3%, and paravalvular abscess in 7.1%. Overall mortality was 14.3%, and infection-specific mortality was 7.1%.

### 3.8. Skin and Soft Tissue Infection Due to Kocuria spp.

The mean age of patients with SSTI caused by *Kocuria* spp. was 56.6 years, and 78.9% were male. The most common clinical presentation of SSTI by *Kocuria* spp. was fever for 28.6% of patients, sepsis for 14.3%, shock for 14.3%, and organ dysfunction for 14.3%. Concomitantly, 10.5% of patients had bacteremia and IE. Overall (and infection-specific) mortality was 5.3%.

### 3.9. Endophthalmitis Due to Kocuria spp.

The mean age of patients with endophthalmitis caused by *Kocuria* spp. was 44 years, and 68.8% were male. Endophthalmitis was exogenous in 93.8%. Mortality was 0%; however, in 81.3% of patients, visual acuity was severely impaired even after the completion of treatment.

### 3.10. Peritonitis Due to Kocuria spp.

The mean age of patients with peritonitis caused by *Kocuria* spp. was 46 years, and 75% were male. The most common underlying condition was peritoneal dialysis due to ESRD in all patients except one (91.7%). The most common clinical presentation included cloudy peritoneal fluid at 100%, abdominal pain at 58.3%, and fever at 41.7%. Overall (and infection-specific) mortality was 8.3%.

## 4. Discussion

This review summarizes the characteristics of all infections caused by *Kocuria* spp. from published studies in the literature that provide adequate data on epidemiology and mortality, as well as on microbiology, clinical characteristics, and treatment. The most common types of infection were bacteremia, skin and soft tissue infection (SSTI), endophthalmitis, infectious endocarditis (IE), and peritonitis. The most commonly identified species was *K. kristinae*; vancomycin was the most widely used antimicrobial. Overall mortality was low.

Only a limited number of articles regarding *Kocuria* infections have been found in the current literature; thus, the definition of exact epidemiological data on these infections remains challenging [42]. In this review, most patients were male, and the mean age was 47 years. Interestingly, most studies on *Kocuria* infections were conducted in Asian countries (53%), while only 27.4% were conducted in European countries. African countries presented the lowest percentage of studies (1.4%). The high percentage of studies in Asian countries could reflect a geographic epidemiology in the young population, given these countries’ young population size. The low percentage of studies in African countries decreases the possibility of a connection between *Kocuria* infections and low-income or poor living conditions. However, due to the limited number of studies and the possibility of underdiagnosis or misdiagnosis worldwide, a safe conclusion has not yet been derived, and more studies are required to elucidate the epidemiology of *Kocuria* infections.

*Kocuria* spp. constitutes part of normal human flora and can be isolated from various environmental niches. More specifically, these pathogens inhabit human skin and mucus membranes, such as the oral cavity, and are usually considered non-pathogenic. So far, microbiology laboratories have treated these bacteria as contaminants, leading to the underdiagnosis of *Kocuria* spp. infections [5]. Infections are most commonly observed in patients with immunosuppression, deformities, those undergoing critical care, or even neonates; however, an increase in *Kocuria* infections has lately been noted in immunocompetent individuals [20]. This pathogen is considered both opportunistic and nosocomial. However, this classification remains challenging; nosocomial infections are more specific in patients undergoing critical care or bearing foreign bodies, such as central venous catheters. In the latter group of patients, biofilm production was suspected to be a potential infection mechanism, although this has not been proved [4].

Regarding predisposing factors, most patients infected by *Kocuria* spp. had a medical history of severe underlying disease, immunosuppression, or indwelling long-term devices [58]. Malignancy constitutes the most common underlying condition noted in the medical history of infected patients. This was even more profound among patients with an SSTI caused by *Kocuria* spp., where more than half of the patients had such a history. About half of the patients were on chemotherapy when the infection was diagnosed. Patients with cancer presented a higher risk for infection, including SSTIs, due to several reasons, such as the immunosuppression associated with cancer, its treatment, anatomical causes that are secondary to the disruption of normal anatomy by the tumor architecture, and the impairment of normal leukocyte function [78,79,80,81]. Moreover, prolonged catheter use and the administration of parenteral nutrition, usually among cancer patients, create an environment that is suitable for bacterial growth. Of note, SSTIs in patients with cancer may carry a higher mortality risk than in patients without oncological history [82,83].

Another common predisposing factor in the studies presented herein was the presence of a central venous catheter (CVC), especially among patients with bacteremia. CVC lines form portals of entry for pathogens to the bloodstream and allow bacterial adherence to the silastic tube. Biofilm development enables bacterial growth and resistance to antibiotic treatment; however, biofilm production in *Kocuria* infections has not yet been confirmed [4]. Indeed, the presence of a CVC significantly increases the risk for bacteremia, especially in specific patient populations, such as in patients with cancer [84,85]. Among the analyzed predisposing risk factors, total parenteral nutrition (TPN) was also relatively frequent in patients infected by *Kocuria* spp., especially in cases of bacteremia. Indeed, an association between TPN and bacteremia has been proposed in the literature, even though a recent systematic review did not firmly confirm this association [86,87,88]. Moreover, this association may be pathogen-specific [86]. 

Several factors that are well-known to be associated with IE were identified in the studies describing patients with IE that were reviewed herein. For example, bad teeth hygiene or recent dental work, congenital heart disease, and intravenous drug users (IVDU) were relatively common among patients with IE by *Kocuria* spp. These factors are associated with IE in the literature [89,90,91]. 

A common predisposing factor identified among patients with peritonitis caused by *Kocuria* spp. was peritoneal dialysis secondary to end-stage renal disease (ESRD). All patients, except one with spontaneous primary peritonitis, presented with a history of ESRD and peritoneal dialysis. Patients with such a history may frequently develop episodes of infection [92]. The microbiology of peritoneal dialysis-associated peritonitis usually involves Gram-positive cocci, most commonly coagulase-negative staphylococci, *S. aureus*, streptococci, and other microorganisms.

The identification of *Kocuria* spp. remains elusive since most microbiology laboratories have limited access to advanced molecular methods. The possibility of the misidentification of *Kocuria* as coagulase-negative *Staphylococcus* based on a Gram stain and catalase and coagulase reactions remains high due to this pathogens’ phenotypic variability [5]. High laboratory suspicion and antimicrobial susceptibility patterns against certain antibiotics can support a prompt diagnosis. For the more accurate identification of this microorganism, advanced microbiological techniques such as 16s rRNA and MALDI-TOF MS are classically needed [5]. In this review, the most common method for pathogen identification was VITEK 2 (BioMerieux Inc., Durham, NC, USA). However, even though there are studies suggesting that VITEK 2 might be adequate for the identification of *Kocuria* spp., there are also reports raising concerns about its ability to identify these microorganisms adequately [93,94]. Thus, ideally, 16s rRNA or MALDI-TOF MS should be used to confirm the identification of *Kocuria* spp. after VITEK 2 identification. Due to these techniques’ low availability and high cost, *Kocuria* spp. could be differentiated from *Micrococcus* spp. and *Staphylococcus* spp. based on their differences in morphology and cultural and biochemical characteristics [4,5,9]. Next Generation Sequencing (NGS) technologies have rapidly evolved over the past years and can potentially identify pathogens by detecting low amounts of DNA or RNA sequences at a low cost [95]. Although NGS could be a diagnostic tool for the prompt and accurate identification of *Kocuria* spp., their role in the diagnosis of this particular pathogen has not been described in the literature; the conduction of further studies is required to provide relevant data.

Currently, no official or widely applied antibiotic susceptibility breakpoint exists for *Kocuria* spp., as no studies have formally assessed the susceptibility profile of this species. Thus, the data in this review are derived from case reports where the antimicrobial susceptibility of the pathogen is mentioned. Antimicrobial resistance patterns of this pathogen present great interest given that it is often underdiagnosed and is also part of the human microbiome. Numerous strains have shown resistance to ampicillin or oxacillin while being susceptible to amoxicillin–clavulanate. This resistance mechanism has yet to be defined.

Nonetheless, a possible explanation would be that a penicillin-binding protein (PBP) mutation affects ampicillin and β lactamase inhibitions and alters cell wall permeability or effluent pumps [1]. Another interesting observation constitutes its resistance to penicillin or oxacillin when combined with susceptibility to ampicillin. Again, no solid explanation has been given in regard to this resistance pattern [1]. Thus, a deeper investigation of these resistance mechanisms and developing specific criteria for interpreting sensitivity assays remains crucial. Based on the data provided by our review, the highest antimicrobial resistance was noted for macrolides and penicillin, where about half of these strains were resistant. The lowest resistance rates corresponded to vancomycin and tetracyclines, while cephalosporin resistance was estimated at 18%.

Protocols regarding the ideal antimicrobial treatment of *Kocuria* spp. infections still need to be clearly defined. According to the data in our review, vancomycin could be the agent of choice for severe infections where pathogen identification is suggestive for *Kocuria* spp. until antimicrobial susceptibility results are available. This aligns with a recent systematic review by Zivkovic et al. that evaluated the antimicrobial treatment of *K. kristinae*. In that review, antimicrobial resistance to vancomycin and linezolid was very low, rendering them an ideal option for empirical therapy after pathogen identification and for as long as the results of antimicrobial susceptibility were pending [96]. In the same systematic review, vancomycin was the most commonly used antimicrobial, followed by cephalosporins, oxacillin, and quinolones. In this review, and in cases where antimicrobial susceptibility results showed resistance to vancomycin (7%), linezolid or quinolones were used as alternative options. In cases where vancomycin resistance was detected, *Kocuria* infections could be treated with any other antibiotic where this pathogen is susceptible, based on the antibiogram.

Similarly, in the present study, where not only *K. kristinae* but all *Kocuria* spp. were evaluated, vancomycin was again the most widely used antibiotic, followed by cephalosporins and quinolones. Treatment duration should depend on the site of infection, and a 10–14-day period of antibiotic administration should be applied in patients with severe infections, such as bacteremia. Moreover, in cases with foreign body-related infections, for example, catheters, the removal of these bodies could prove beneficial [1].

The mortality rate of infections caused by *Kocuria* spp. was relatively low, differing among infection types. For example, in IE, mortality was higher than other infections. The mortality of IE was comparable to those in studies with IE caused by other bacteria [97,98,99]. In endophthalmitis, mortality was 0%; however, the persistent, significant visual loss rate after treatment was exceptionally high. This is in line with other studies reporting data on endophthalmitis caused by other pathogens, where visual acuity is also significantly impaired for the majority of affected patients [100,101].

This study had some limitations, mainly related to its narrative nature. The literature search may have been incomplete for all studies providing adequate data on epidemiology and mortality, and due to the search strategy, some studies may have been missed. Our study included only case reports and case series whose credibility depends on accurate record keeping. Moreover, since in most studies, molecular identification with 16s rRNA or MALDI-TOF MS was not performed, there is a possibility that misidentification has occurred in some cases. Finally, some of the presented data were missing in some studies; thus, statistical analysis was conducted using only the available data, allowing us to present only the results deriving from studies with available information.

## 5. Conclusions

This study describes the epidemiology, clinical characteristics, microbiology, antimicrobial susceptibility, treatment, and outcomes of *Kocuria* spp. infections, highlighting the potential pathogenic nature of this microorganism. *K. kristinae* was the most commonly isolated species, bacteremia was the most common infection, and antimicrobial resistance to macrolides and penicillin was very high. At the same time, although validated therapeutic guidelines do not exist, vancomycin and tetracyclines appeared to have the lowest resistance. Vancomycin was the most widely used antimicrobial. The empirical antimicrobial treatment of *Kocuria* spp. infections should include vancomycin as long as antimicrobial susceptibility results are pending. The infection outcome mainly depends on the type of infection, and its severity is associated with the patient’s immune state. Clinical and microbiological suspicion, early diagnosis, and appropriate antibiotic therapy remain the mainstay for managing *Kocuria* infections.

## Figures and Tables

**Figure 1 microorganisms-11-02362-f001:**
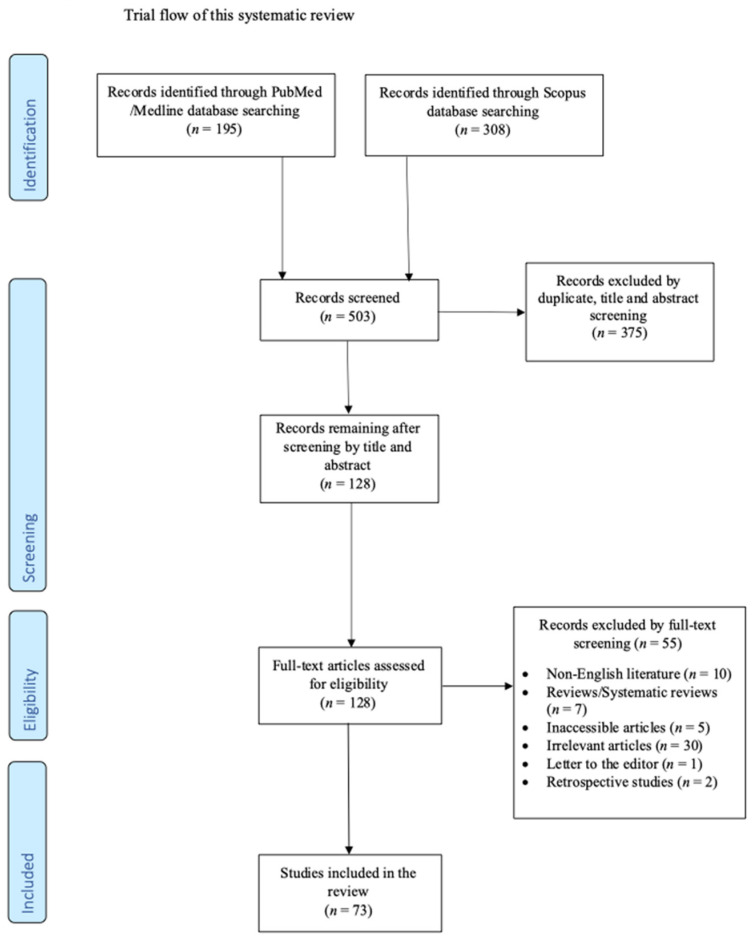
Flow diagram of study inclusion.

**Figure 2 microorganisms-11-02362-f002:**
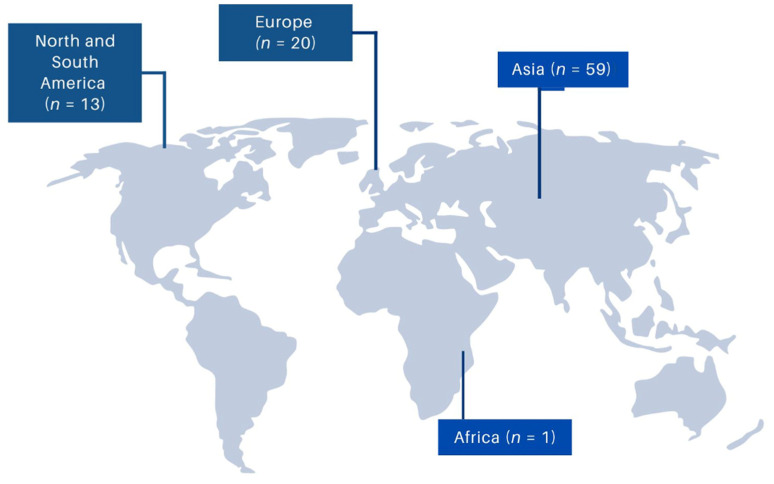
Geographical distribution of *Kocuria* spp. infections worldwide.

**Table 1 microorganisms-11-02362-t001:** Characteristics of the different types of infections by *Kocuria* spp.

Characteristic *	Bacteremia ** (*n* = 37)	Infective Endocarditis (*n* = 14)	SSTI (*n* = 19)	Endophthalmitis (*n* = 16)	Peritonitis (*n* = 12)
Age, mean (SD)	42.5 (29.2)	58.2 (22)	56.6 (15.6)	44 (21.8)	46 (31.4)
Male, *n* (%)	22/36 (61.1)	12 (85.7)	15 (78.9)	11 (68.8)	9 (75)
Active malignancy, *n* (%)	4 (10.8)	0 (0)	13 (68.4)	0 (0)	1 (8.3)
On chemotherapy, *n* (%)	3 (8.1)	0 (0)	10 (55.6)	0 (0)	0 (0)
CVC, *n* (%)	16/35 (45.7)	2/13 (15.4)	0 (0)	0 (0)	2 (18.2)
TPN, *n* (%)	13/36 (36.1)	1/13 (7.7)	0 (0)	0 (0)	0 (0)
Bad teeth hygiene or recent dental work, *n* (%)	3 (8.1)	3 (21.4)	0 (0)	0 (0)	0 (0)
IVDU, *n* (%)	2/35 (5.7)	2/13 (15.4)	0 (0)	0 (0)	0 (0)
Congenital heart disease, *n* (%)	2 (5.4)	1 (7.1)	0 (0)	0 (0)	0 (0)
ESRD on PD, *n* (%)	1 (2.7)	0 (0)	0 (0)	0 (0)	11 (91.2)
Microbiology					
*K. kristinae*, *n* (%)	15 (40.5)	6 (42.9)	13 (72.2)	6 (37.5)	3 (25)
*K rosae*, *n* (%)	9 (24.3)	5 (35.7)	1 (5.6)	5 (31.3)	2 (16.7)
*K. rhizophila*, *n* (%)	4 (10.8)	1 (7/1)	0 (0)	2 (12.5)	1 (8.3)
*K. varians*, *n* (%)	4 (10.8)	0 (0)	0 (0)	3 (18.8)	1 (8.3)
*K. marina*, *n* (%)	2 (5.4)	0 (0)	0 (0)	0 (0)	3 (25)
*Kocuria* spp., *n* (%)	2 (5.4)	2 (14.3)	0 (0)	0 (0)	0 (0)
*K. koreensis*, *n* (%)	0 (0)	0 (0)	1 (5.6)	0 (0)	0 (0)
*K. palustris*, *n* (%)	0 (0)	0 (0)	1 (5.6)	0 (0)	0 (0)
*K. salsicia*, *n* (%)	1 (2.7)	0 (0)	0 (0)	0 (0)	1 (8.3)
*K. arsenatis*, *n* (%)	0 (0)	0 (0)	1 (5.6)	0 (0)	1 (8.3)
*K. massiliensis*, *n* (%)	0 (0)	0 (0)	1 (5.6)	0 (0)	0 (0)
Clinical characteristics					
Fever, *n* (%)	30 (81.1)	11 (78.6)	2/7 (27.6)	0 (0)	5 (41.7)
Sepsis, *n* (%)	12 (32.4)	4 (28.6)	1/7 (14.3)	0 (0)	0 (0)
Organ dysfunction, *n* (%)	9/34 (26.5)	7 (50)	1/7 (14.3)	0 (0)	0 (0)
Shock, *n* (%)	5/36 (13.9)	4 (28.6)	1/7 (14.3)	0 (0)	0 (0)
Embolic phenomena, *n* (%)	6/36 (16.7)	5 (35.7)	1/7 (14.3)	0 (0)	0 (0)
Outcomes					
Overall mortality, *n* (%)	3 (8.1)	2 (14.3)	1 (5.3)	0 (0)	1 (8.3)
Infection-related mortality, *n* (%)	2 (5.4)	1 (7.1)	1 (5.3)	0 (0)	1 (8.3)
Significant vision loss, *n* (%)	NA	NA	NA	13 (81.3)	NA

CVC: central venous disease, ESRD: end-stage renal disease; IVDU: intravenous drug user, NA: not applicable, PD: peritoneal dialysis, SD: standard deviation, SSTI: skin and soft tissue infection, TPN: total parenteral nutrition, * denominator is the total number of patients except if otherwise mentioned, ** cases of bacteremia include the cases of infective endocarditis.

## Data Availability

The data presented in this study are available on request from the corresponding author.

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
