# Peer review of "Kocuria Species Infections in Humans—A Narrative Review"

_microorganisms, 2023, doi:10.3390/microorganisms11092362_

Round 1

Reviewer 1 Report

Ziogou A et al reviewed the Kocuria infections in humans reporting data on epidemiology and mortality. The Authors retrieved 73 studies providing data from 102 patients.

They concluded that our knowledge about many aspects of Kocuria infection is not defined and we need more data.

The review is well-designed and developed.

I have a few comments to make regarding the Results and the Discussion sections.

Results: In the first subchapter, the Authors reported that  73 studies provided data from 102 patients. It would be very interesting to also provide a cartoon showing the countries, and the specific number of cases of Kocuria infections, to have a visual distribution worldwide of the epidemiological data.

Discussion: as reported in Results, the mean age of infection was 47 years. This value is low, indicating a high incidence of infection in the pediatric and young population. However, does it also depend on the geographic epidemiology of the infection (most countries are in Asia, are they low-income countries with young populations?), or does this phenomenon simply reflect the high incidence of infection in young people in developed countries? Please comment on this issue in the Discussion

good 

Author Response

Ziogou A et al reviewed the Kocuria infections in humans reporting data on epidemiology and mortality. The Authors retrieved 73 studies providing data from 102 patients.

They concluded that our knowledge about many aspects of Kocuria infection is not defined and we need more data.

The review is well-designed and developed.

I have a few comments to make regarding the Results and the Discussion sections.

Results: In the first subchapter, the Authors reported that 73 studies provided data from 102 patients. It would be very interesting to also provide a cartoon showing the countries, and the specific number of cases of Kocuria infections, to have a visual distribution worldwide of the epidemiological data.

Response: Thank you for the comment. We created an image representing the number of studies on Kocuria spp. infections published worldwide as an additional way to provide a clear understanding of the pathogen’s distribution.

Discussion: as reported in Results, the mean age of infection was 47 years. This value is low, indicating a high incidence of infection in the pediatric and young population. However, does it also depend on the geographic epidemiology of the infection (most countries are in Asia, are they low-income countries with young populations?), or does this phenomenon simply reflect the high incidence of infection in young people in developed countries? Please comment on this issue in the Discussion.

Response: Thank you for your comment. This information would be interesting and helpful to the readers. Unfortunately, there is limited data on Kocuria infections published in the literature, and most articles on this pathogen are case reports. As a result, the authors could not reach a safe conclusion regarding this epidemiological information. However, the second paragraph that has been added in the discussion section provides more information on this topic.

Reviewer 2 Report

General comments

=============

I appreciated the opportunity to peer-review your work on the narrative review for Kocuria species infections in humans. A primary suggestion is to revamp the overall structure of the manuscript to ensure a coherent and systematic flow of content. This will aid the readers in comprehending the central research question, the methods you've adopted, and the resultant findings. To facilitate this understanding, I propose that the authors offer a concise outline or a summarization of the entire content in the introduction. Such an inclusion would offer readers a clear roadmap of the article and can substantially enhance the reading experience.

Specific comments

=============

Major comments

---------------------

General comments elaboration:

-----------------------------

1. The abstract section mentions, “The most frequently isolated species was K. kristinae, and antimicrobial resistance was lower for vancomycin and tetracyclines. Vancomycin, cephalosporins, and quinolones were the most commonly used antimicrobials.” For clarity and quantification, it would be beneficial if you could include the specific number or percentages of cases related to these findings.

2. In the introduction, it would be prudent to detail what existing knowledge gaps or unknown areas in the field prompted the undertaking of this narrative review. This will help readers understand the rationale behind the study.

3. For the methods section, the process of sourcing articles from PubMed and Scopus requires elucidation. Please provide details like who conducted these searches, the total number of articles retrieved, those that were excluded, and the reasons for their exclusion. Furthermore, adopting a study flow diagram, perhaps something inspired by the PRISMA flow, could provide readers with a visual representation of the selection process.

4. The sentence “Among them, 39 were conducted in Asia, 20 in Europe, 13 in North 59 and South America, and one in Africa” needs clarity. Are these numbers referring to individual studies or patients?

5. When mentioning that “TPN was also relatively frequent in patients with infection by Kocuria spp., especially in cases of bacteremia”, it would be beneficial to specify the comparison population. The term "relatively" necessitates a reference point.

6. In the discussion section, there seems to be a significant emphasis on previous literature not directly related to Kocuria infections. I suggest narrowing down the discussion to primarily focus on the results of this narrative review. Content discussing unrelated Kocuria infections might be better placed in the introduction or removed to maintain focus.

7. There's an evident contradiction in the manuscript. While you've discussed the absence of antibiotic susceptibility breakpoints for Kocuria spp., you've also presented data on antibiotic sensitivity. Kindly address and resolve this discrepancy.

8. Any narrative review inevitably encounters some data gaps. Please elaborate on your approach to handle missing data or information from the studies you reviewed.

9. The manuscript highlights vancomycin as the standard choice of treatment for Kocuria infections. However, with a reported 7% vancomycin-resistant Kocuria infection (as mentioned in line 89), it's pivotal to provide guidance or recommendations on treating such cases.

---------------------

Minor comments

---------------------

10. For the benefit of readers unfamiliar with the term, consider providing a brief description or elaboration of “PubMed.”

11. Before introducing abbreviations such as CSF and ICD, it's standard practice to spell out the terms in full on their first appearance. This ensures that even readers unfamiliar with the abbreviations can grasp their meaning.

Please refer the previous "Quality of English Language"

Author Response

General comments

=============

I appreciated the opportunity to peer-review your work on the narrative review for Kocuria species infections in humans. A primary suggestion is to revamp the overall structure of the manuscript to ensure a coherent and systematic flow of content. This will aid the readers in comprehending the central research question, the methods you've adopted, and the resultant findings. To facilitate this understanding, I propose that the authors offer a concise outline or a summarization of the entire content in the introduction. Such an inclusion would offer readers a clear roadmap of the article and can substantially enhance the reading experience.

Response: Thank you for your recommendation. We attempted to describe the central research question in lines 50-55 of the Introduction section. The text “Advanced microbiological techniques, such as 16rRNA, are required for an accurate identification [5]. Currently, there exist no official treatment guidelines for Kocuria infections; most cases are initially treated empirically until antimicrobial susceptibility results are available. Mortality of these infections was generally low and mainly depended on the site of the infection as well as the patient’s comorbidities. “ was also added in the introduction in lines 44-49 to provide a better outline of the article's content.

Specific comments

=============

Major comments

---------------------

General comments elaboration:

-----------------------------

  1. The abstract section mentions, “The most frequently isolated species was K. kristinae, and antimicrobial resistance was lower for vancomycin and tetracyclines. Vancomycin, cephalosporins, and quinolones were the most commonly used antimicrobials.” For clarity and quantification, it would be beneficial if you could include the specific number or percentages of cases related to these findings.

Response: Thank you for your comment. The required percentages of cases related to these findings have been included in lines 20,21 and 22 of the Abstract section.

  1. In the introduction, it would be prudent to detail what existing knowledge gaps or unknown areas in the field prompted the undertaking of this narrative review. This will help readers understand the rationale behind the study.

Response: Thank you for your comment. The sentence “This review constitutes an attempt to fill certain knowledge gaps, including predisposing factors or treatment options, and to enhance the limited data available in the literature for this pathogen.” has been added in lines 53-55 of the introduction section to provide examples of unknown areas and help readers understand the authors’ rationale.

  1. For the methods section, the process of sourcing articles from PubMed and Scopus requires elucidation. Please provide details like who conducted these searches, the total number of articles retrieved, those that were excluded, and the reasons for their exclusion. Furthermore, adopting a study flow diagram, perhaps something inspired by the PRISMA flow, could provide readers with a visual representation of the selection process.

Response: Thank you for your comment. The Materials and Methods section has been fully modified accordingly in lines 57-77.

  1. The sentence “Among them, 39 were conducted in Asia, 20 in Europe, 13 in North and South America, and one in Africa” needs clarity. Are these numbers referring to individual studies or patients?

Response: Thank you for your comment. Since these numbers correspond to individual studies, the term “studies” has been added next to each number to clarify this issue in lines 84 and 85 in the Results section.

  1. When mentioning that “TPN was also relatively frequent in patients with infection by Kocuria spp., especially in cases of bacteremia”, it would be beneficial to specify the comparison population. The term "relatively" necessitates a reference point.

Response: Thank you for your comment. The phrase “Among the analyzed predisposing risk factors” has been added in line 313 to explain that the term “relatively” is used to show TPN’s frequency compared to other risk factors.

  1. In the discussion section, there seems to be a significant emphasis on previous literature not directly related to Kocuria infections. I suggest narrowing down the discussion to primarily focus on the results of this narrative review. Content discussing unrelated Kocuria infections might be better placed in the introduction or removed to maintain focus.

Response: Thank you for your comment. We narrowed down the discussion section by removing the three following sentences: 1. “For example, TPN has been identified as an independent factor associated with the development of candidemia in patients in the intensive care unit [89].”, 2. “IE is classically caused by Gram-positive microorganisms, such as Staphylococcus aureus, streptococci, and enterococci; however, other organisms may infrequently cause IE [93–95].”, 3.” Other pathogens, such as Gram-negative microorganisms like Enterobacterales, may also be implicated, especially in polymicrobial infections [99-101].” In the discussion section.

  1. There's an evident contradiction in the manuscript. While you've discussed the absence of antibiotic susceptibility breakpoints for Kocuria spp., you've also presented data on antibiotic sensitivity. Kindly address and resolve this discrepancy.

Response: Thank you for your comment. Data presented in this review derive from the authors’ research of the current literature according to certain inclusion criteria.  By mentioning the absence of antibiotic susceptibility breakpoints, the authors’ aim is to show that this particular topic has not been well studied yet, and no official data/ guidelines have been set yet. To clarify this issue, the phrase “no antibiotic susceptibility breakpoints exist for Kocuria spp.” has been modified to “no official and widely applied antibiotic susceptibility breakpoints exist for Kocuria spp.” in line 371 and the phrase “the only available data” has been altered to “the data in this review” in line 373 of the discussion section.

  1. Any narrative review inevitably encounters some data gaps. Please elaborate on your approach to handle missing data or information from the studies you reviewed.

Response: Thank you for your comment. The phrase “statistical analysis was conducted using only the available data” has been added to line 426 of the discussion section, and the phrase “allowing us to show only the data among studies with available information.” has been modified to “allowing us to present only the results deriving from studies with available information.” in lines 426-7. Moreover, the sentence “Our study included only case reports and case series whose credibility depends on accurate record keeping has been added in lines 422-3 to explain the reason for certain data gaps further.

  1. The manuscript highlights vancomycin as the standard choice of treatment for Kocuria infections. However, with a reported 7% vancomycin-resistant Kocuria infection (as mentioned in line 89), it's pivotal to provide guidance or recommendations on treating such cases.

Response: Thank you for your comment. The following text has been added in the discussion section in lines 398-402: “In this review, in cases where antimicrobial susceptibility results showed resistance to vancomycin (7%), linezolid or quinolones were used as alternative options. In cases where vancomycin resistance is detected, Kocuria infections could be treated with any other antibiotic where the pathogen is susceptible, based on the antibiogram.”

---------------------

Minor comments

---------------------

  1. For the benefit of readers unfamiliar with the term, consider providing a brief description or elaboration of “PubMed.”

Response: Thank you for your comment. The term “database” has been added in lines 15 and 16 of the Abstract section.

  1. Before introducing abbreviations such as CSF and ICD, it's standard practice to spell out the terms in full on their first appearance. This ensures that even readers unfamiliar with the abbreviations can grasp their meaning.

Response:  Thank you for your observation. The abbreviations CSF and ICD have been spelled out in full in the results section/subsection 3.3 in lines 155-6. 

Round 2

Reviewer 1 Report

I have no further comments

Good